# Documenting Aggression, Dominance and the Impacts of Visitor Interaction on Galápagos Tortoises (*Chelonoidis nigra*) in a Zoo Setting

**DOI:** 10.3390/ani10040699

**Published:** 2020-04-17

**Authors:** Laura Freeland, Charlotte Ellis, Christopher J. Michaels

**Affiliations:** 1Royal Veterinary College, University of London, Royal College Street, London NW1 0TU, UK; 2Zoological Society of London, Regent’s Park, London NW1 4RY, UK; charlotte.ellis@zsl.org (C.E.); christoper.michaels@zsl.org (C.J.M.)

**Keywords:** behaviour, Elo rating, fixed action pattern, welfare, zoo

## Abstract

**Simple Summary:**

Understanding the mechanisms by which welfare can be optimised in zoos is essential for improving standards of animal care. Using scan-sampling methods, assessments of group hierarchy and direct observations, the behaviour of Galápagos tortoises (*Chelonoidis nigra*) was assessed in relation to public interaction. We observed that head movements and height were of importance in aggressive interactions. We found that the presence of, and type of, visitors (keepers, vets or public) within the enclosure influenced behaviour in this species, with increasing levels of aggression and activity. We suggest that when visitors initiate finching, a behaviour in which a tortoise will stretch its body upwards to allow birds to remove ectoparasite in the wild, they negatively influence tortoise welfare. We suggest that careful management of public interaction with this species can improve welfare.

**Abstract:**

Ensuring high levels of welfare is imperative for modern zoos, but such organisations must also engage visitors in order to successfully spread awareness and raise conservation funds. It is therefore important to understand the responses of animals to visitor interaction to optimise welfare. Often, the opportunity to interact with humans may be enriching for animals, but in other contexts, this interaction may have negative welfare effects. We observed captive female Galápagos giant tortoises (*Chelonoidis nigra*) to describe aggressive interactions, characterize hierarchy using Elo ratings and assess the impact of visitor interactions. Elo ratings indicated that one individual was dominant over two equally ranked subordinates; aggressive interactions are discussed in this context. We detected significant effects of the presence of visitors and visitor type (keepers, vets or public) within the enclosure on aggression and activity. We suggest that previous miscategorisation of a natural behaviour (the finch response) as an operantly conditioned behaviour, rather than a fixed action pattern, may have triggered aggression. We then document changes made to the management of the animals to mitigate the impacts discovered. This work highlights the importance of empirical evidence in determining optimal management strategies for zoo animals with regards to public interactions and animal welfare.

## 1. Introduction

Direct interaction with zoo animals may influence more people to visit a zoo, engage in conservation and even pay additional fees to participate with animals, which can support conservation work [1]. The effects of human interaction on captive animals, however, is hugely varied and often unknown [2]. At present, most data concerning human–animal interaction in zoos are derived from studies of mammals, which suggest that people may act as enrichment, a source of stress, or have neutral impact [2]. Despite 21% of WAZA zoos allowing the public to touch live reptiles [3], few if any data exist on how these interactions affect animal welfare [3,4,5]. Understanding the impacts that visitors may have on captive animals is vital in enabling zoos to engage and educate visitors, whilst promoting positive animal welfare [6,7].

Galápagos giant tortoises (*Chelonoidis nigra*) are charismatic reptiles that present little risk to human safety and so are often used for interaction with the public. This interaction may occur when zoo visitors, as well as the husbandry and veterinary staff who work with all animals, engage in husbandry as part of experiences sold by the zoo. These experiences are intended to provide enrichment to the animals, raise funds for conservation and further educate the public. Members of the public may also be able to touch tortoises, as well as undertake shell scrubbing or initiating the finch response [4]. The finch response is a natural behaviour in which a tortoise will stretch out its limbs and head to allow birds such as finches or mocking-birds to remove ectoparasites [8,9]. This can be initiated by scratching the neck or legs of the tortoise, resulting in the tortoise performing the same outstretched pose [9]. It is believed that finching giant tortoises may act as enrichment through direct interaction, stimulation of natural behaviours or simply by offering variability in daily routines [4]. Finching is also regarded as a reinforcer for use in behavioural management, and ‘finching’ tortoises has been reported as a positive reinforcer in operant conditioning programmes to facilitate veterinary procedures [10]. However, the impacts of such visitor interactions on giant tortoises have not been assessed empirically. Given the tendency to underestimate welfare issues in reptiles [11], it is important that management decisions regarding animal–human interactions are made in an evidence-based way to ensure that animal welfare is promoted and negative impacts are avoided.

A possible sign of compromised welfare described in other species is aggressive behaviour [12,13]. Elevated levels of aggression in captivity may result from fear, lack of enrichment, or injury [14]. Intraspecific aggression is a natural behaviour in giant tortoises and has been described both in wild settings and in captivity in mixed sex groups. In this species, aggression consists of individuals raising themselves up, exhibiting horizontal and vertical head movements, biting and hissing [8,15,16]. Female–female aggression, however, has not been formally described. Elevated aggression may lead to injuries from biting, and in captivity, this may result in reduced animal welfare if conditions driving increased aggression are not addressed.

Another reason for aggression within the group is in response to competition or to establish a dominance hierarchy [17]. Hierarchy in tortoises is believed to be established and maintained via displays of aggression and regulating access to resources and mates [15]. Understanding group hierarchy may offer an explanation as to why certain individuals within a group display more aggressive behaviour than others [18]. A method for determining group hierarchy in animals is the Elo rating method, created by Arpad Elo, which enables ranks to be determined and updated continuously as interactions occur [19].

We observed the behaviour of a group of three *C. nigra* in a zoo setting, which were involved in routine direct interaction with keepers, veterinarians and members of the public. The aims of this study were to (1) characterise aggressive female–female behaviours, (2) assess if a dominance hierarchy exists within the group, as little is known of dominance hierarchies amongst female Galápagos tortoises, and (3) assess if the interaction with people influences aggression in Galápagos tortoises, in order to inform management decisions regarding these animals, and to quantify results of any changes made.

## 2. Methods

### 2.1. Study Subjects

We observed three 24-year-old female *Chelonoidis nigra* at ZSL London Zoo, UK. These animals had the Species360 Zoological Information Management System (ZIMS; a global database for zoo animal data) local identifiers 7605, E0273 and E0274, which will be used to refer to the animals herein. All three animals were captive bred at Zurich Zoo and were siblings. At the time of the study, E0273 and E0274 had been housed at ZSL London Zoo for six years and 7605 for eight years after a history of being housed in other European zoos with other giant tortoises. 

The Galápagos tortoise indoor enclosure was 4.64 m by 14.84 m at its maximum dimensions (Figure 1). A pond was located near to the viewing window, and a fence near the centre of the enclosure allowed an individual animal to be separated from conspecifics if necessary. A single, large basking area comprising two conjoined equipment rigs (each using one 150W BaskZone heater (SunSwitch, UK) and two banks of eight 54W 12% UVB T5 fluorescent lamps (Arcadia, UK) in a reflective housing (Growth Technology, UK)) provided a source of UVB (UVindex of 4.9–5.6 within the bask area) and heat (37–45 °C in the bask area). The main enclosure was temperature controlled with air conditioning units between 22 °C at night and 30 °C during the day. The animals did not have access to the large outside paddock as this study occurred during winter time when external temperatures were too low to permit outside access.

### 2.2. Study Design

Three time-lapse cameras (Plotwatcher Pro Day 6 Outdoors) were installed in the Galápagos tortoise house. The view of the cameras covered most of the indoor area (see Figure 1). Each camera was set to take a photograph at five second intervals from 7 am until 6 pm daily, from the 28th January 2019 to the 25th March 2019. Five second intervals were chosen to maximise the amount of data collected, while allowing for most behaviours to still be observed. Stop-motion footage was played back for coding using GameFinder (Day 6 Outdoors) at 4× playback speed. To aid identification of each tortoise, several spots of non-toxic coloured nail varnish were temporarily painted on the carapace of each tortoise. All behavioural data were collected from these images.

Days throughout the study were grouped into either visit days or control days; control days were days in which only keepers entered the enclosure, whereas on visit days there were additional people entering the enclosure. Visitors allowed within the enclosure (henceforth simply ‘visitors’) were grouped into four types: keeper only (KO), VIP encounter (VIP), vet visit (VV) or keeper for a day (KFAD, including both Junior and Adult versions). Keeper husbandry occurred as normal on VIP, VV and KFAD days. At the time of the study, KFAD was a paid, structured experience in which up to five individuals were allowed to touch and finch the tortoises, accompanied by a member of zoo staff who was not a member of the tortoise husbandry team. The age range for KFAD varied widely and included both children and adults. VIP was a non-paid experience and usually consists of smaller groups of adults in which finching and interacting with the tortoises was also allowed; this was accompanied by a member of the regular tortoise husbandry team. KFAD and VIP visits were intended to provide enrichment to the animals, while facilitating conservation education and conservation fundraising. VV often involved carrying out routine veterinary checks and treatments; finching was often used to separate the tortoises and ensure they stood still for veterinary examination. Zoo visitors viewing the tortoises from outside of the enclosure were not recorded during this study. 

### 2.3. Aggression Levels on Visit and Control Days

Preliminary observations were carried out by observing two days of footage in order to describe the full behavioural repertoire of female Galápagos tortoises in captivity (Table 1). We refer to behaviours that we presumed to be aggressive (see Table 1) simply as ‘aggression’ henceforth. On all study days, the total number of aggressive interactions were recorded from 7 am until 6 pm, as well as the total number of visitors, including keepers, and the type of visitors entering the enclosure. These observations were used to describe aggressive behaviours.

### 2.4. Aggression and Activity before, during and after Visit

Scan sampling of footage was used to assess behaviours every two minutes before, during and after a non-keeper visit occurred using an ethogram (Table 1). We coded all behaviours as state behaviours due to their slow nature and classed as active or non-active (see Table 1) to assess if interaction with people may alter activity levels. One researcher (LF) analysed all footage. We designated behaviours to ‘before’, ‘during’ and ‘after’ human interaction. The ‘during’ sample window varied according to the requirements of the interaction and the ‘before’ and ‘after’ phases were set to 16 min (the median interaction duration either side of the interaction).

### 2.5. Hierarchy Assessment

We used the Elo Rating method to assess the rank of each Galápagos tortoise, as well as changes in rank within the group [19,20]. This method determines the rank of individuals over time, taking into account the likelihood of individuals winning based on past interactions. Furthermore, this method also shows the magnitude of difference between individuals, allowing interpretations to be made about how closely ranked individuals within a group are [18]. For each aggressive interaction observed throughout the study, the ‘actor’, ‘recipient’, ‘winner’ and ‘loser’ were recorded based on the outcome of the interaction. The ‘actor’ was the individual that first approached another tortoise or raised its head in close proximity to another tortoise, hence, beginning a sequence of aggressive interactions. The ‘winner’ of an aggressive interaction was the individual that did not get displaced, or whose head remained raised with no further aggression. The ‘loser’ was the displaced individual or the individual that retracted their head into their shell with no further interaction occurring. 

We measured the maximum head height of each tortoise from the ground to the top of the head using a tape measure, because past studies indicate that height may play a role in dominance in chelonians [16,21]. We used food to encourage the tortoise to stretch to their maximum height.

### 2.6. Statistical Analyses

Statistical analyses were undertaken using R-3.6.1 [22]. The significance level for all tests was set at 0.05. Shapiro–Wilk tests were run to assess the normality of the data, and as LOG transformation failed to normalise the data, non-parametric tests were used. We compared total number of aggressive interactions on control days compared to visit days using a Mann–Whitney U test. Due to the small sample size of tortoises (n = 3), we treated days as the experimental unit and accepted pseudoreplication and, therefore, inapplicability of results to the wider population of *Chelonoidis nigra* as a limitation of the study [23]. We also compared visit length on visitor days with number of aggressive interactions on that day using a Spearman’s Rho.

We assessed whether the number of visitors or type of visitors influenced aggression with a Spearman’s Rank test and a Kruskall–Wallis test, respectively. We used a Friedman test to compare the proportion of time engaged in aggression and the proportion of time spent active before, during and after visitation. We also tested for differences in time spent out of sight between ‘before’, ‘during’ and ‘after’ samples using a Friedman test. Dunn’s multiple comparison tests and Tukey range tests were also used to compare visitor types and interaction period, respectively.

We calculated Elo Ratings using the ‘Elorating’ package v0.46.8 in the R statistical environment [22,24]. The Elo Ratings start value was set to 1000 and the maximum change in ratings was set at a constant (k = 100) [22,24]. Any variation in the intensity of aggression was not recorded, and therefore, each interaction was weighted equally.

### 2.7. Post-Study Mitigation

Following analysis of the results from the study (see below), finching the tortoises by the public was discontinued from 22nd March 2019. Veterinary records from August 2018 to July 2019 were reviewed to calculate the number of bite wounds per month caused by aggression, and the severity of wounds (the proportion of wounds per month requiring veterinary treatment) during historic management (1st August 2018 to 21st March 2019), during implementation of change in management (22nd March 2018 to 30th April 2019) and post implementation (1st May 2019 to 31st July 2019). Although some wounds required veterinary attention, none was severe enough to pose a substantial health problem for the animals in question. 

## 3. Results

### 3.1. Description of Aggressive Interactions

Aggression occurred on all days recorded (n = 47). Over the study period, the mean number of aggressive interactions for the whole group was 6.638 per day (SD = 2.785, n = 279). Aggressive interactions were usually initiated when one tortoise approached another with its head raised, but the initiation of aggressive interactions occasionally occurred via a head raise when eating (Figure 2a). One tortoise sometimes bit another with little or no prior signs of aggression, resulting in the head retreat and displacement of the recipient tortoise, or progression into a display of dominance. The dominance display consisted of a series of head raising and air biting, resulting in the submission of one individual, discontinuation by both individuals, or more biting (Figure 2b–e). Air biting was a behaviour where a tortoise would raise its head and open and close its mouth as if biting the air. This series of interactions could continue for significant periods of time, lasting up to two hours in some cases. Biting to the face, neck and legs was observed. Once bitten, a tortoise would rapidly withdraw its head into its carapace. 

### 3.2. Hierarchy

Throughout the study, 279 aggressive interactions were recorded in total. All three tortoises were observed initiating and being the recipient of aggressive interactions, with no one tortoise winning or losing all fights (Table 2, Figure 3). 

### 3.3. Aggression, Activity and Visitor Interaction

Days in which the camera SD cards or batteries were changed were excluded from analysis. Days in which there was more than one visitor type were also excluded to prevent bias. This resulted in a total of 47 days being used for analysis. 

No significant difference was found between the number of aggressive interactions on visitor days (n = 29) in comparison to control days (n = 18) (W_46_ = 179.5, *p* = 0.074). However, the median number of aggressive interactions was greater on visitor days (median = 7) than for control days (median = 6). A significant positive correlation was observed between the number of aggressive interactions and the number of visitors (Rs_46_ = 0.334, *p* = 0.022) (Figure 4a). No significance was found between the duration of visit and number of aggressive interactions on that day (Rs_28_ = −0.152, *p* = 0.430). 

A significant effect of visitor type on aggression was found (H_3_ = 16.026, *p* = 0.001) and a Dunn’s multiple comparison test showed significantly higher levels of aggression on KFTD days than VV days (*p* = 0.003) and KO days (*p* = 0.011) (Figure 4b).

There was no significant difference in time spent out of sight between interaction periods (*X*^2^_2_ = 0.207, *p* = 0.902). We found a significant effect of the time before, during and after visit on aggression (*X*^2^_2_ = 8.149, *p* = 0.017) (Figure 4c). The mean number of aggressive interactions before, during and after visit were 1.000, 0.586 and 1.828. A Tukey’s range test showed significantly higher percentages of aggression after visit in comparison to during visit (*p* = 0.019). Furthermore, a significant change in activity levels before, during and after aggression was found (*X*^2^_2_ = 21.586, *p* < 0.001) (Figure 4d). A Tukey’s range test indicated significantly higher activity before visit in comparison to during (*p* < 0.001) and after visit (*p* < 0.001).

### 3.4. Post-Study Mitigation Response

Records indicated a reduction in wounding following discontinuation of finching (prior to change = 5.188 injuries per month, during change = 12.403 injuries per month, post-change = 3.33 injuries per month). Records also indicate that post-change, a smaller proportion of wounds required veterinary care, suggesting a reduction in the severity of wounding (prior to change = 72.5% of injuries requiring veterinary care, during change = 56.25%, post-change = 10%).

## 4. Discussion

By observing a captive group of Galápagos tortoises, we were able to characterise the aggressive actions and sequences of aggression that occur between females of this species. We reveal that human presence within the enclosure, and the type of interaction involved, influenced the activity levels and aggression. Furthermore, via use of Elo ratings, we describe the hierarchical relationships that exist within this group of female Galápagos tortoises and show that they are likely characterised by size and aggression.

The context in which aggression occurred varied within our study; in the presence of resources (for example, food or access through a gateway), when encountering another tortoise while locomoting, and after targeted approaches by one animal towards another without any apparent competition for resource. However, no matter the context, the aggressive interactions followed similar sequences (Figure 2). Visual cues involved in the dominance display (see Figure 2), including head raising and air-biting, often resulted in displacement without physical aggression. This is likely a mechanism by which fighting prowess may be gauged, and interactions resolved without the risks of injury and energy expenditure involved in actual fights [25]. Encounters were initiated and resolved in several different ways, even between the same individuals, but with a limited range of sequences possible (Figure 2). If physical aggression did occur, both the head and legs were targets of bites during aggressive interaction, as reported in wild male *C. nigra* by Auffenberg (1977) [15], but bites to the neck were also commonly observed in our study. 

Groups of captive animals will usually form temporally stable dominance hierarchies that are associated with reduced frequency and/or intensity of aggression following an establishment period where these behaviours may increase [15]. E0273 and E0274 were housed together for six years prior to the start of this study, yet the continuation of aggression suggests a well-established and stable dominance hierarchy may not have formed [26]. Frequencies of aggression in wild females are unknown, but Schafer and O’Niel (1983) [16] reported fewer than 0.14 aggressive interactions per hour at San Diego Zoo, which is considerably lower than the 0.55 aggressive interactions per hour reported here. However, this may have been due to the higher stocking densities at ZSL London Zoo. This high frequency of aggression, alongside frequent changes in Elo rating ranks, suggests that a stable dominance hierarchy may not have formed in the ZSL group [26]. Our Elo rating results do, however, indicate that 7605 was the most dominant animal in the group, at least during the study period. This animal initiated the least aggression, which may reflect that as a dominant there is a reduced need to fight for resources, or that the relative risk of losing an aggressive interaction when already in the dominant position [27]. 

Relative height is an important determinant of hierarchy in tortoises, with taller animals being more dominant [16,21]. 7605, with the highest Elo rating, was also the tallest. In contrast, E0274, who scored below 7605 and above E0273, measured the shortest of the three. The height difference between E0274 and E0273 was less than that between either animal and 7605; the low variation in height between animals may explain the small difference in, and fluctuation in rank order of, Elo ratings in this study. 7605 also received the most aggression of all the animals in this study, which may reflect frequent challenges to her position from conspecifics, who were only slightly shorter than her. Moreover, 7605 was housed at the zoo for around two years before the arrival of E0274 and E0273; this may have also contributed to her dominant status as she was resident when the other animals arrived. Gerlach (2003) [28] suggests that variation in the size of tortoises in a group may promote the formation of a stable dominance hierarchy and associated reduced aggression, and therefore, the similarity in size of the three tortoises in this study may have promoted further aggression. 7605 may have initiated fewer aggressive interactions, as she had more to lose than to gain from aggression, being the dominant individual [27].

We showed that the number and type of visitors entering the Galápagos tortoise enclosure influenced aggressive behaviour between individual animals in the group. Although aggression and resultant wounding is considered part of the natural behavioural repertoire for this species [15], high rates of aggression and wounding may be indicative of and a cause of sub-optimal welfare in captive animals [12]. Although we did not find a significant difference in aggression between visitor and control days, we did find a significant positive correlation between numbers of visitors and aggression. Aggression therefore appears to increase proportionately with increasing visitor numbers on a continuum as opposed to an all-or-nothing response to the presence visitors regardless of their number. The correlation between aggression and visitor numbers may be the result of the increased perceived threat from a larger visitor group size or higher volumes of sound being produced from larger groups [6]. Although it is known that sound levels can affect animal behaviour, there is little evidence that it can induce aggression, especially in reptiles [29,30].

Aggression was significantly lower during the KFAD visits, as tortoises were immobilised by the finch response, but once the visit was over and animals left the immobile state, aggression occurred for significantly greater percentages of the time than prior to KFAD visitor presence. Having the choice to participate in enrichment or training is a key determinant of animal welfare [31] and should always be promoted by an enrichment strategy [32]. Indeed, removing choice over participation from animals may result in aggression [33]. Eliciting the finch response in giant tortoises has been historically understood as being an operantly conditioned behaviour and that exhibition of the behaviour is a matter of choice for the animal [10]. However, in the absence of extreme perturbing circumstances, giant tortoises appear to invariably respond to tactile stimuli by finching in captivity. This is not typical of operantly conditioned behaviours; rather, we suggest that the behaviour is a fixed action pattern (FAP). FAPs are distinctive, stereotyped (in the sense of their structure, rather than welfare implications) behaviours that are triggered by a simple external stimulus called an innate releasing mechanism (IRM) [34]. Once triggered, the behaviour will be performed in its entirety [35]. The IRM in the wild is the physical and vocal display of the finch or the sensation of it removing ectoparasites from the tortoise [8,36]. If finching is an FAP, then in captivity, the IRM would be tactile stimulation by the keeper. If this is the case, then approaching tortoises and initiating the behaviour without allowing the animals to exercise choice over whether to participate would result in the removal of choice from the animal. As well as increased aggression as a response to reduction in behavioural choice, activity levels also increased post visit. Higher levels of activity may increase the likelihood of encountering another individual and therefore heightened the chance of an aggressive interaction. Moreover, bites from a non-finching individual to a finching individual were commonly observed. Further work to investigate the integration of finching with choice provision should be explored.

Finching and agonistic displays also have overlapping postural characters, namely the head raised and outstretched [36]. The finch position may be mistaken as a display of dominance by nearby animals, therefore, triggering aggression or submission. Where veterinary visits involved finching tortoises, the response was only initiated after animals had been deliberately separated and therefore less likely to mistake finching for aggressive posturing. This, along with relative familiarity of vets compared with members of the public on KFAD visits, could explain the reduced aggression during veterinary visits compared with KFAD encounters. The combination of increased aggression due to the removal of choice, increased activity leading to higher encounter rates, and the similarity between finching and aggressive postures may have been important in driving increased aggression levels due to visitor interactions.

Our data also suggest that discontinuation of elicitation of the finch response as part of a visitor experience substantially reduced both the rate of wounding through aggression and also the severity of wounds that were inflicted on animals by conspecifics. Although the timespan of data collection for this element of the study was limited, these results further support the role of mistakenly treating the finch response as an operantly conditioned behaviour, rather than an FAP, in driving increased aggression. The increase in wounding and severity of wounds seen during the brief adjustment period is likely the result of changes to the animals’ routines perturbing normal behavioural patterns [37]. Overall, by adjusting management strategies according to data, we were able to mitigate welfare implications of humans interacting with tortoises in their enclosure while maintaining positive aspects of human interaction.

The small sample size was a limitation of this study, and due to our statistical approach, differences between giant tortoise collections such as habitat or husbandry, it is not possible to say whether these results would be reflected across the captive population of this species. Further research using multiple giant tortoise collections and variable group sizes would aid our understanding of this species’ response to human interaction and help to further understand aggression and hierarchy.

## 5. Conclusions

We documented in detail aggressive interactions between captive female tortoises and showed that direct interactions between visitors and giant tortoises can result in reduced welfare for the animals in question through its effects on levels of aggression. This is likely at least partly caused by misunderstanding of the nature of a fixed action pattern behaviour (finching) and miscategorising it as an operantly conditioned behaviour, which may drive increased aggression through reduction in behavioural choice. Information of this sort is important in determining how zoos manage captive animals to maximise animal welfare and conservation impact and demonstrates the importance of making conservative management choices based on evidence.

## Figures and Tables

**Figure 1 animals-10-00699-f001:**
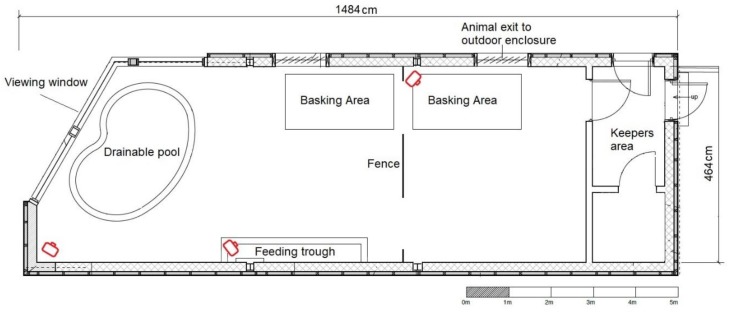
A map of the indoor Galápagos tortoise enclosure at ZSL London Zoo. The red symbols indicate the location of the cameras.

**Figure 2 animals-10-00699-f002:**
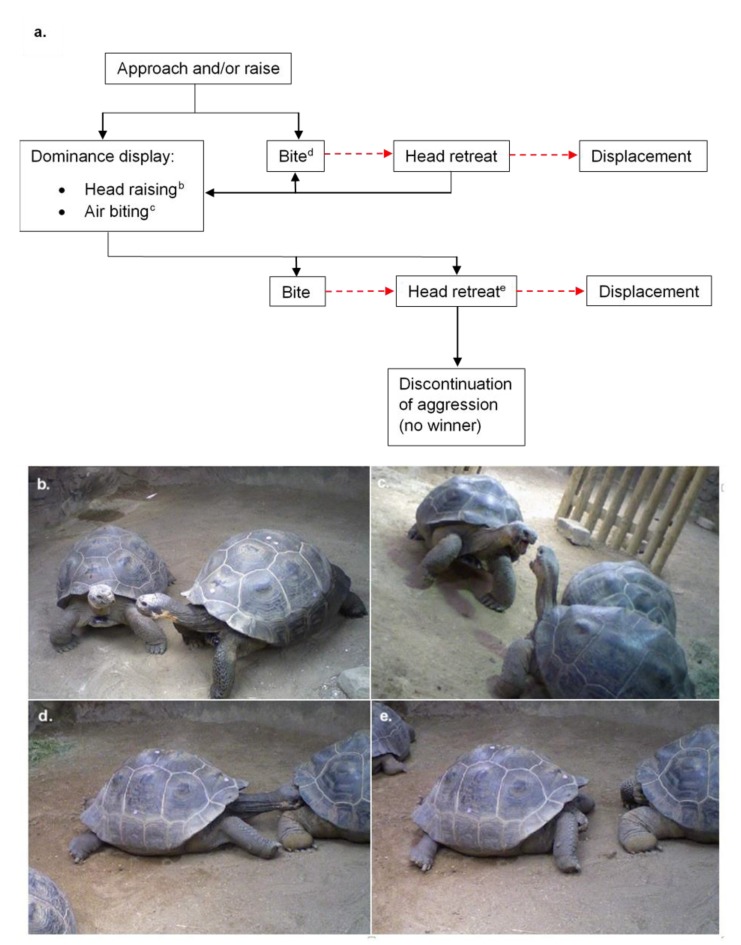
(**a**) A flow sequence of the aggressive behaviours that may occur between two Galápagos tortoises. Each interaction may be undertaken by either or both tortoises, with the exception that the tortoise that is bitten will be the one to retreat and move away (indicated in a red dashed line); (**b**) head raising; (**c**) air biting; (**d**) biting; (**e**) head retreat.

**Figure 3 animals-10-00699-f003:**
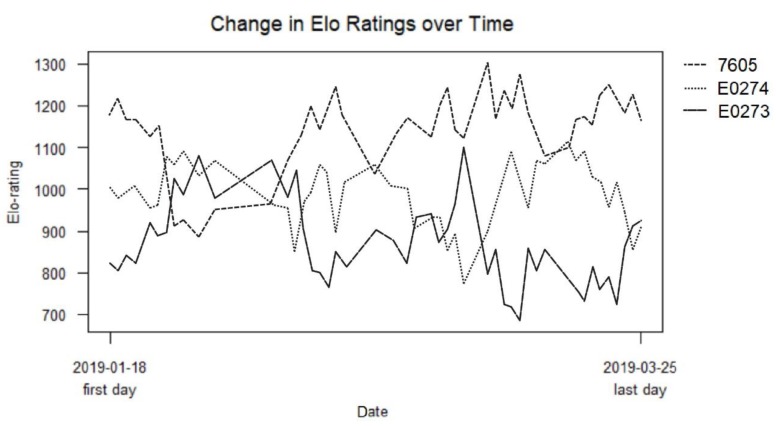
Elo ratings of the aggressive interactions between 7605, E0274 and E0273 over the course of the study. The first point for each line represents the rank differential that has been generated from one day of interactions.

**Figure 4 animals-10-00699-f004:**
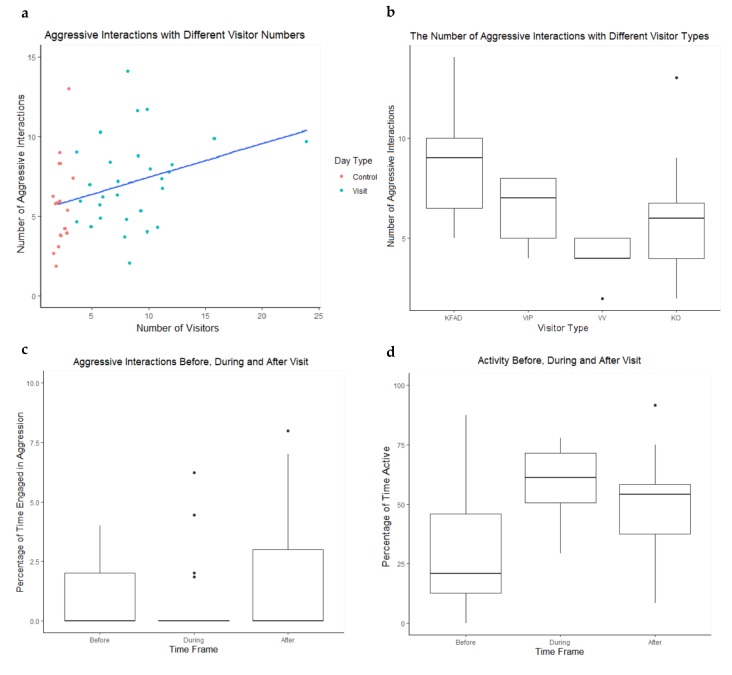
(**a**) The relationship between the number of visitors per day and the number of aggressive interactions that occurred between Galápagos tortoises on control and visitor days. Red dots indicate control days and blue dots indicate visitor days. (**b**) A box whisker plot showing the number of aggressive interactions per day with different visitor types, KFAD = Keeper for a Day, VIP = VIP Encounter, VV = Vet Visit, KO = Keeper Only. (**c**) A box whisker plot showing the percentage of time spent engaged in aggressive interactions before, during and after visit times. (**d**) A box whisker plot showing the percentage of time spent active before, during and after visit times. The bold line of the boxplot indicates the mean, the box indicates the upper and lower interquartile ranges, the whiskers indicate the highest and lowest observation and the points represent outliers.

**Table 1 animals-10-00699-t001:** An ethogram describing the different behaviours exhibited by Galápagos tortoises and whether they are classed as active or non-active behaviours.

Behaviour	Group	Description
Sleeping	Non-active	Flat on floor, still, head down
Resting	Non-active	Flat on floor, some movement of head and/or legs
Walking	Active	Locomotion, excluding the below behaviours
Standing	Active	Raised above the floor, stationary
Eating	Active	Consuming food or water
Finching	Active	Fully raised head and body, while stationary
Aggressive Interaction	Active	Raising head higher than resting level or air biting in close proximity to another tortoise, or biting another tortoise
Non- Aggressive Interaction	Active	Movement of head in close proximity to another tortoise, not including actions included within ‘aggressive interaction’
Bathing	Active	Part or all of the tortoise in the pond
Other	N/A	Any behaviour not described by other definitions
Out of Sight	N/A	The tortoise is not visible from any camera

**Table 2 animals-10-00699-t002:** The final Elo rating scores, number of times actor and recipient of aggression, number of times winner and loser of aggression, total number of aggression interactions and height of the three Galápagos tortoises. The higher the Elo rating, the more dominant the individual.

Individual	Final Elo Rating	No. of Times Actor of Aggression	No. of Times Recipient of Aggression	No. of Times Winner of Aggressive Interaction	No. of Times Loser of Aggressive Interaction	Total No. of Aggressive Interactions	Height (cm)
7605	1165	69	119	146	42	188	86.5
E0274	924	104	92	55	141	196	84
E0273	911	106	68	78	96	174	85

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
