# Peer review of "Documenting Aggression, Dominance and the Impacts of Visitor Interaction on Galápagos Tortoises (Chelonoidis nigra) in a Zoo Setting"

_animals, 2020, doi:10.3390/ani10040699_

Round 1

Reviewer 1 Report

This paper is of interest to readers and generally well written. There are a few minor issues for consideration.

There are mistakes with punctuation, e.g. Line 265 - no comma after 'height'.

Introduction: I think more is needed here. For example, some of what is in the discussion should be introduced here to be picked up later in discussion.

  • line 52 - do all visitors participate in husbandry procedures and is finching considered husbandry?
  • line 55 is is 'mimicked' or 'initiated'?
  • line 80 the 'between females' is not needed as already says 'aggressive female-female behaviours'

Methodology:

  • Does Elo stand for anything and what is the origin of this rating? The introduction of Elo could occur in Introduction rather than here. This section just says how it was done in this study
  • Duration of the aggression activity and the severity seem like important measures so please explain no attempt was made to include these.
  • Other studies have considered visual cues with respect to aggression - why was that not done here?

Results

  • Figure 2 - (d) is not included in diagram in (a)
  • What is the duration of change. It seems like the chance was just a cessation of something so please more clearly explain the results listed in Line 236.
  • You showed a correlation between number of visitors and type of visitors but what about length of visit?

Discussion

  • Bottom of first paragraph - Are you talking about previous research or to your research?
  • The information on FAP & Operant conditioning should be introduced in Introduction

Author Response

Word file attached

Reviewer 2 Report

Review of 761848 - Documenting aggression, dominance and the impacts of visitor interaction on Galapagos tortoises (Chelonoidis nigra) in a zoo setting

This is an interesting paper that focuses on a common welfare issue in zoos (human-animal interactions) in an understudied taxonomic group. While the sample size is small and there are questions about whether the statistics are appropriate (detailed below), I think this is a valuable contribution to the field of zoo animal behavior and welfare, and hope the authors can improve the report to prepare it for publication.

The writing style needs to be improved throughout. Many sentences should be re-written to be more direct and concise. Clarity is often obscured by convoluted sentence structures. In single paragraphs there are a mix of past, current and future verb tenses. I suggest sending the manuscript to a colleague or editor who focuses on academic writing; this issues go beyond copy editing. (The first sentence the introduction is a good example of the troubles throughout; the semi-colon is used incorrectly, it is not clear to whom the idea is supposedly compelling, and the number of ideas combined in the single sentence is overwhelming and difficult to parse.)

In the simple summary, it is not clear what is meant by “hierarchical assessments” (line 11)

In the abstract, it is not clear what is meant by “we mapped the behavioral interactions” (line 24)

In simple summary and abstract, it is not evident what is meant by “type” of visitor (line 14, line 29)

Introduction

Line 46 should include reference to the limited published literature on reptiles.

Line 52: “carers” should be “caretakers”

Line 50 - 52: Unclear what is meant here, tortoises may be subject to visitor interactions by caretakers, vets or visitors? Perhaps “visitor interaction” should be “human interaction” in line 50 if this is meant to be a more general human-animal interaction discussion? (Also relevant in line 60.) Related, do you mean to say that visitors may be responsible for husbandry practices? This may need more explanation.

The introduction is currently a bit slim. I think reference to other ambassador/interactive studies could be useful before narrowing the focus to replies. Some references of interest may be D’Cruze et al 2019 reporting on widespread popularity of interaction programs globally, Baird et al., 2016 reporting on a multi-institutional study evaluating welfare of three species in visitor interaction programs, Saiyed et al., 2019 reporting on penguin behavior / welfare in a visitor interaction program, Orban et al., 2016 reporting on behavior / welfare of giraffes in a visitor feeding program. There is also a recent dissertation by Acaralp-Rehnberg that is relevant. Full refs provided at the end.

The synopsis of aims at the end of the introduction (lines 79-83) is very helpful & clear.

Methods:

Line 104 explains photographs were taken, but line 106 and 109 reference footage that was played back (implying video).

Section 2.2: This section should contain an overview of the study design, e.g., that there are visit and control days, and that there were before/during/after visit conditions. I find myself trying to piece together the study design as I read through the subsequent paragraphs.

How was time not visible accounted for? It seems raw counts were used, this should be corrected by visible time per subject.

Was inter-observer reliability determined? If not, a subset of videos / should be recoded by an observer blind to the hypotheses and results at this point. 

Line 128: I suggest replacing “visitors” with a unique term for people entering the exhibit. The term “visitors” may be confusing to readers, since it usually suggests people viewing from outside the exhibit.

Line 129: Meaning of this sentence not clear: “Aggression was described by observing each of these behaviors.”

Section 2.5: One key feature (and valuable feature) of the Elo Rating method is that it does not assign only rank order but also magnitude of differences between ranks (see also a recent zoo paper using Elo method in Woods et al., 2019 - either that paper or references within may be useful). This aspect should be included in the description of the Elo method.

Section 2.7: The Elo package should be included here rather than above. This section should be paragraph style, as it is currently written it ends with a list of statements.

Section 2.8: This is an interesting portion of the study, but it is confusing to be presented here when the reader does not yet know the results of the study. I wonder if it is possible to move the information about the change in management practices, and the evaluation of that change, to a new Study 2?

Section 3.1: Suggest retitling this section to “Description of aggressive interactions” or something else to be clear this is the first aim, to characterize female-female aggression in this species.

Figure 2a: This is a clear presentation, but does need some clarification of whether this is the only set of sequences, represents typical sequences, etc. Could you say something to the effect of xx% of documented interactions followed one of these sequences?

Figure 3: Shouldn’t all three individuals start at 1000?

Section 3.3: It looks like the Mann-Whitney degrees of freedom relates to the number of days, not the number of tortoises. The sample size here is 3, and the unit of analysis should be the individual to avoid pseudoreplicaiton and maintain independence of data. Please check Mann-Whitney U based on individual. I’m not sure that is possible with an N of 3, may have to result to reporting individual results rather than statistical analyses here.  Likewise, the correlation reported on lines 213-214 does not seem to take into account that there are repeated measures from individuals, again compromising independence. This will take more thought with the data, but a more appropriate approach may be a logistic mixed effects model considering individual as a random effect, visitor count as a predictor, and the outcome variable being whether or not aggression occurred that day.  For the remainder of statistical tests int his section, I am curious whether data are independent or whether all results were pooled across subjects, leading to pseudoreplication/lack of independence. I understand the challenge of small sample sizes and support the authors in providing individual data summaries/visualizations without statistics if that is the most appropriate for their data set.

Discussion: Suggest starting out with a brief recap of the goals of the study to reorient the reader.

Line 263: This interpretation seems less likely than the need for aggressive interactions being lower for the most dominant - animals tend to avoid aggression if possible and if this individual has priority of access to resources due to her dominant status, why would she be motivated to engage in aggression that is always risky (not for losing status, but for incurring wounds)?

Line 278: Be careful when interpreting your results, we don’t know about causal factors in this design, just associations

Line 287: Is sound known to be aversive? This seems logical but needs explanation/unpacking here.

Line 297: “Choice in participation…, and should always be promoted by an enrichment strategy.” This sentence does not make sense to me.

Lines 299-309: I find this to be an interesting and somewhat unique discussion. Makes me wonder whether you would hypothesize that the triggering of any FAP then lead to frustration due to a lack of choice/control? Or is there some valence to consider? Not sure you need to integrate this here.

Conclusion: Well summarized.

Some minor comments (not exhaustive):

line 11: scan-sapling should be scan-sampling

line 63: period missing

line 69: Hatt 2008 not formatted per journal style

line 99: suggest more specific word than “plan” - perhaps map, layout, schematic drawing, etc.

line 132: classes should be classed

line 175: none “was” should be none “were”

Acaralp-Rehnberg, L. K. (2019). Human-animal interaction in the modern zoo: Live animal encounter programs and associated effects on animal welfare (Doctoral dissertation). LINK to full thesis.

Baird, B. A., Kuhar, C. W., Lukas, K. E., Amendolagine, L. A., Fuller, G. A., Nemet, J., ... & Schook, M. W. (2016). Program animal welfare: Using behavioral and physiological measures to assess the well-being of animals used for education programs in zoos. Applied Animal Behaviour Science176, 150-162.

D’Cruze, N., Khan, S., Carder, G., Megson, D., Coulthard, E., Norrey, J., & Groves, G. (2019). A Global Review of Animal–Visitor Interactions in Modern Zoos and Aquariums and Their Implications for Wild Animal Welfare. Animals9(6), 332. Link to full article.

Orban, D. A., Siegford, J. M., & Snider, R. J. (2016). Effects of guest feeding programs on captive giraffe behavior. Zoo Biology35(2), 157-166.

Saiyed, S. T., Hopper, L. M., & Cronin, K. A. (2019). Evaluating the behavior and temperament of African penguins in a non-contact animal encounter program. Animals9(6), 326. LINK to full article.

Woods, J. M., Ross, S. R., & Cronin, K. A. (2019). The Social Rank of Zoo-Housed Japanese Macaques is a Predictor of Visitor-Directed Aggression. Animals9(6), 316. LINK to full article.

Reviewer 3 Report

This is a really interesting paper from a class of vertebrates frequently overlooked for such behavioural analyses. It has had immediate and justifiable impact in changing management practices of the Galápagos tortoise at ZSL London Zoo, and I have no doubt will inform practices worldwide. However, I was left unsatisfied with the arguments presented regarding tortoise aggression and Operant Conditioning/ Fixed Action Patterns in this paper. I feel that although an admirable amount of data from a small sample of animals has been presented well and analysed thoroughly, the authors have sold themselves very short with the interpretations of the evidence they provide. It is on this basis that I recommend major revision before publication of this otherwise excellent study. I will outline my reasons below after I address the cosmetic typographical and presentation issues I identified:

Typos etc:
Line 10: underlining of space rather than section title
Line 11: scan-sapling

Line 30: "We suggest that miscategorisation of a natural behaviour (the finch response) as an operantly conditioned behaviour, rather than a Fixed Action Pattern, may have created a trigger for aggression linked to visitor-animal interactions." Needs to be stated more clearly - the tortoises are miscategorising, or the observers? [This is a complex statement, and I flagged it to come back to after reading the main text. Indeed this is one of the key areas that wasn't addressed for me through the manuscript. My initial critical thought on reading this was "Could the aggressors be initiating attacks opportunistically at others immobilized during the finch response?" 

Line 52: "Mimicking" is bad choice of term here and below. Suggest 'evoked' or 'initiated' instead, e.g. "It may also occur by direct interaction in the form of shell scrubbing or BY EVOKING the finching response..." 

Line 55: "This can be mimicked by scratching the neck or legs of the tortoise, resulting in the tortoise performing the same outstretched pose [7]"

Line 87: "...animals have the ZIMS codes 7605, E0274 and E0274" (??)

Figure 1: Seems to have incorrect dimensional measurements by an order of magnitude as the scale does not correspond with the distances given (i.e., 148.4m x 46.4m shown in figure as cm)

*Apart from the alternative approach for considering the social agonistic behaviours which I cover below, I was intrigued that the comparison with data from San Diego Zoo (0.14 aggressive interactions per hour) didn't address stocking density as a possible covariate in the expression of agonistic behaviour in this species. Human-derived stressors are a clear cofactor in the expression of agonistic tortoises here. Although the paper explains the necessity to overwinter in a comparatively smaller area, I was surprised nonetheless that there wasn't some additional brief comparison made regarding the relative stocking densities between the two reports (e.g, I roughly calculate 1 tortoise/19 square meters at ZSL (0.55 aggressive interactions per hour), does that compare well with the San Diego situation - is there an inverse relationship, if any?)

Presentation of Figure 4b was a little problematic for me, as the VV category data are presented in the middle of the other conditions suggesting a continuum of human-related stressors - however the text indicates that the turtles were assessed in visual isolation. Thus I feel that this category should be presented to the side (i.e., change places with the KO condition, perhaps a brief description provided in the figure caption to assist reader).

Rationale for Major Revision recommendation

Tortoises (Testudo hermanni) have recently found to posses lateralized visual processing for social cues - in alignment with other vertebrate species studied (Ref. Sovrano, V.A.; Quaresmini, C.; Stancher, G. Tortoises in front of mirrors: Brain asymmetries and lateralized behaviours in the tortoise (Testudo hermanni). Behavioural Brain Research 2018, 352, 183–186). You present a figure image of a left-sided agonistic encounter in two of your females (cf. Fig 2d,2e). While this example is illustrative only (and n=1), should the pattern of left-eye dominance for directing social information be demonstrated throughout your photographic records - then this truly would be a significant study of major scientific importance.

I have not worked with tortoises but I can appreciate how their general behavioural repertoire and evolutionary history may appeal to traditional mechanistic explanations of Fixed Action Patterns and Innate Releasing Mechanisms. However, do these models of behaviour adequately account for the possibility of premeditated behaviours? Could aggressor tortoises be intentionally and opportunistically taking advantage of the tonic immobility (finching responses) of their opponent to safely inflict a wound, or indeed reciprocate an earlier attack from the same individual? This is where I think the Elorating approach tends to pool the encounter data without addressing structural sequences (if any) of bouts of agonistic interaction underlying the social in your animals. Similarly, the absence of lateralized cognitive processing of social visual cues in this study would also be an important contrast to the pattern of examples identified across vertebrate species.

There are several avenues that could easily be employed to expand the importance of this paper, and improve the interpretations presented so far. Current evidence from the literature would suggest that ritualized, agonistic responses (i.e., head elevation, air biting) would tend to be initiated by tortoises using predominately their right-visual field to display their dominance. By contrast, actualized agonistic responses (bites causing injury) may tend to be more opportunistic or spontaneous in nature, and be directed within the predominant left-side visual field of the aggressor. This is just speculation. What is important however is to determine whether such encounters are actually random and in accord with FAP and IRM - type explanations, or whether there is structure within the data that fully describes the social relationships between these fascinating animals.

In conclusion, I am completely convinced that clear evidence has been provided to indicate that human-associated stressors are significantly and positively correlated with reduced welfare outcomes for these tortoises. I am however unconvinced with the arguments presented that Chelonoidis nigra tortoises potentially mistake in their opponents a posture associated with finching, with that of ritualized dominance posturing in the species. The opportunity to expand the analysis of aggressive encounters in these captive tortoises could include any veterinary records showing/discounting 'sidedness' in bite injuries sustained, any detail of delayed reciprocity of attacks from a finched tortoise injured during a human visitation, towards the previous attacker in the post-visit period. In reassessing the encounter data using such an approach, the authors may be able to make not only a significant contribution to both the understanding of Galápagos tortoise welfare and behavioural requirements, but also to the burgeoning field of lateralized cognitive processing in vertebrates.

Thank you for an extremely stimulating study,

Kind Regards

Reviewer 4 Report

Documenting aggression, dominance and the impacts of visitor interaction on Galápagos tortoises (Chelonoidis nigra) in a zoo setting

I would like to thank the authors for this nicely written and well-structured manuscript. Despite the limitation of the small sample size, which is well highlighted by the authors, this investigation within a zoo setting is, in my view very relevant especially in relation to the overall goal of ameliorating animals living conditions. Introduction, results and discussion sections are fine. The statistical approach is appropriate considering the low sample size (n=3). The methods section can be improved in clarity and details. First of all, I would put all the contents of the methods in one section instead of dispersing them throughout. For example, in the study design I would include also the number of days and the exclusion criteria (line 207-209) used.

Line 11 replace with “sampling”.

Line 21-24 I would probably split in two sentences.

Line 24-27 this sentence is not clear probably because too long or because too many information are presented. Same issue is visible from line 30-32. I would probably revise the abstract overall especially when the methods and results are presented.

Line 49 It might be worth to consider citing the paper “A “How-To” Guide for Designing Judgment Bias Studies to Assess Captive Animal Welfare” by Emily J. Bethellhttps://www.tandfonline.com/doi/full/10.1080/10888705.2015.1075833?casa_token=vGI4R6YFzvMAAAAA%3Afz5tb06BCNxZtpFobYTffQpAyZNx4JGPFQRvlOC2W-q0K7LbYBZZNHqFbJ-2Lnc5YVfe-vN5JBvQ

Line 57 “these” is in this context referring specifically to the finch response?

Line 70 how did the authors define stress here?

Line 74 -79 revise these sentences. Aim 3 also requires some revision because it is not very clear.

Line 102 I do not understand the sampling methods used to score the behaviours listed in the ethogram. The camera continuously recorded for 13 h per day and every 5 sec the camera captured a photo (9360 shots). How did the authors process the coding?

Line 114 If age is later discussed, then the actual age range should be provided; otherwise consider removing it.

Line 125 how long for did the authors perform the preliminary observations?

Line 133 the way in which the scan sampling footage was performed is not clear. The authors selected a 16 min time window (960 sec) and I assume that within this time window, 480 shots were scored. Based on this, the state (duration ?) of behaviours was reconstructed although with several 5 sec missing. Is that true? Also, why did the authors select state rather than events? I think this section is not matching the general scoring illustrated in line 102. 

Line 151-154 probably, this should be moved in statistical analyses.

Line 155 to 157 I assume height was taken because of its link with the dominance status. Please provide a justification for including this measure in the text.

Line 162 Why did the authors choose to perform the Mann-Whitney U test rather than a Kruskall-Wallis test (4 types of visitors: Keeper, VIP, VV, KFAD)? I would first investigate if aggressive behaviour is affected by the type of visitors and then carry out Mann-Whitney U tests corrected for multiple comparisons.   

Line 168: post study mitigation; the decision to finch the tortoise was taken as a result of this study if I have understood it correctly. If so, it would be appropriate to move this in the discussion. This is very nice and the fact that the management took into consideration the findings of this study is something that authors should highlight more.

Line 187 did the authors record the 2 h long aggressive interaction or is it something available in literature? It appears quite a long time.

Line 188 in addition to the total number of days, I would also specify the total number of aggressive behaviours recorded ( n = 279).

Line 210 in this case the authors have combined all types of visitors and compared it against control. This is correct, however I am not sure it is needed considering that the authors have performed the correlation and after the Kruskall-Wallis test.

Figure 3 I am not sure it is appropriate to use red and green because some people with impaired vision might have difficulties with it (unless a grey scale will be used in the final draft).

Figure 4 Is the box plot correct? It looks like there are more outliers than subjects. If the data were plotted using R, some preliminary calculations (mean and sd) for all the subjects in the three time points in which aggression occurred might be needed, unless the authors wanted to show some different aspect in which case the figure legend should be changed accordingly.

Line 244 fights instead of combat?

Line 294 In the original paper the authors described how to use a spontaneous behaviour (the neck extension in reaction to a stimulus) to perform some veterinary care which does not have anything to do with operant conditioning. The authors in the original paper just used this spontaneous response (reflex or FAP) to familiarise the tortoise with the needle (desensitisation) and collect the blood sample.

Line 305 I am not convinced that the FAP (or reflex) of neck extension could be considered as outcome of a choice made by the animal (choice which if frustrated would trigger aggression, based on the current interpretation of findings). I would argue instead that the behaviour of neck extension is a spontaneous behaviour emitted by the animal in response to tactile stimulation, which might serve an evolutionary purpose. If the authors wanted to prove the point that aggression is triggered by the frustration of a choice, then they should consider using classical or operant conditioning and measure the preference of the animal to be touched. Also, including multimodal measures (e.g. cortisol response, physiological reactivity such as cardiac activity) before, during and after being touched would provide additional useful information to infer the quality of the perception of the stimulation.

Round 2

Reviewer 2 Report

The authors provided a revision that largely satisfies my earlier concerns and I think the manuscript has been greatly improved. Limitations and weaknesses remain (sample size limiting statistical options, lack of second observer), but they have been clearly acknowledged by the authors and cannot easily remedied. I think this work will make a valuable contribution to the field, especially given the limited knowledge on the impact of visitor interactions on zoo animal welfare and the paucity of data on reptile welfare. I have only minor comments below. 

First, a follow up comment related to the responses:

Figure 3’s legend has become more confusing with the additional statement, although I appreciate it was meant to clarify. What about “The first point for each line represents the rank differential that has been generated from one day of interactions.” ?

In addition, in the revised version:

Line 268: I think it is more appropriate to say that via the use of Elo ratings, you describe the current hierarchical relationship between these three females, and show that the relationships are related to size. (The aggression is how you defined the Elo ratings, so by mentioning aggression this becomes a circular statement. Related, this is a study of three females with questionable generalizability, as the authors note.)

Finally, there are several instances where the writing can be improved and I detail them below:

Line 11: if you are talking about optimizing (changing, improving) welfare, then I think “maintaining” is not the appropriate word here but rather you should refer to “advancing” or “improving” standards of care.

Line 13: is -> was, and edits to past tense throughout (line 14: “are” to “were”, 15: “influences” to “influenced”)

Line 14: “visitors” to “visitors inside the habitat”

Line 16: sentence starting “We suggest” is long and confusing, what about “We suggest that when visitors mimic finching, a behavior in which a tortoise stretches its boy to allow birds to remove ectoparasites, they negatively influence tortoise welfare.”

Line 19: replace “can address associated welfare issues” with “can improve welfare.”

Line 24: Shorten sentence and remove unnecessary detail, suggest: “We observed captive female Galapagos giant tortoises (Chelonoidis nigra) to describe aggressive interactions, characterize their hierarchy using Elo ratings, and assess the impact of visitor interactions.”

Line 29: “visitors” to “visitors inside the habitat”

Line 31: Shorten and simplify: …”may have triggered aggression.”

Line 39: Difficult to parse. What about: Direct interaction with zoo animals may influence more people to visit a zoo, engage in conservation, and provide financial support for zoos and their missions.

Line 42: “are derived from”

Line 46 “is vital in enabling…” to “is essential if zoos wish to engage visitors while promoting positive animal welfare.” Otherwise you are building in assumptions about how ambassador programs sustain conservation goals, which is not established.

Line 73: suggest replacing “determine” with “establish”

Line 89: remove comma after “three”

Line 90: rework sentence to be more direct, also you may need to reference ZIMS

Line 121: the visitors aren’t behind the scenes, correct? just in the habitat? (e.g., public could see them)?

Line 137: suggest deleting “full” - two days will not provide a full repertoire, likely.

Line 141: Suggest removing circularity and rephrasing to “ These observations were used to describe aggressive behavior.”

Line 145: “on the minute” not important here, would just say every two minutes for clarity

Throughout: standardize number of significant digits

Line 271: “one another” should be “another”

Line 339: Sentence starting on this line confusing, suggest rework into two sentences

Reviewer 3 Report

I thank the authors for their consideration of my previous comments, and they have adequately rebutted my suggestions regarding furthering the analysis to examine the data for lateralized, and possibly opportunistic/premeditated behaviours adequately. 

I did note however the term "mimicked" had slipped by me previously, in the Abstract, line 18. Could the authors change this please as it is incorrectly used in the context of the sentence, as in the earlier examples I provided.

I could have been more explicit with my earlier comment (now line 91)"...local identifiers 7605, E0274 and E0274 which..." E0274 is listed twice.

I thank the authors for incorporating the suggested changes and justifying why other significant changes are beyond the remit of the current data set. I feel that given the above minor changes, the paper is now acceptable for publication with Animals MDPI.
